# Adaptive Thinking in Cities: Urban Continuity within Built Environments

Hana Morel [1,*] and Brenda Denise Dorpalen [2]

1    MOLA (Museum of London Archaeology), London N1 7ED, UK
2    Environment Agency, Economics, Appraisal and Research, London SW1P 4DF, UK
*    Correspondence: hmorel@mola.org.uk

**Abstract:** Adaptive reuse of built heritage is increasingly critical for reasons of sustainability, particularly in urban spaces. With increasing pressures for building and housing, the building and construction industry will likely continue to contribute 39% of all carbon emissions in the world, with operational emissions accounting for 28%. Further demolition, urban renewal and rebuilding not only obstruct pathways to decarbonisation but create shocks that disrupt and displace communities. We argue that it is essential to support built heritage beyond conventional urban renewal approaches and to position it as a critical community-based asset that can leverage local knowledge and promote a sustainable and more circular economy. However, such an agenda must acknowledge the challenges of adopting new innovative practices that can reduce strain on current material and social resources. This paper situates adaptive reuse as critical in strategies to reuse existing building stock and other broader sustainability movements, framing it from an economic angle. A case study approach explores adaptive reuse interventions and how these might be extended to other areas.

**Keywords:** historic buildings; adaptive reuse; sustainability; circular economy; economic growth

## 1. Introduction

Sustainability involves the preservation of people's well-being across generations [1]. It considers social, environmental, cultural and economic dimensions [2] in the pursuit of meeting every person's needs while preserving the living world on which we all depend [3]. At the crossroads between the economic, social, cultural and economic dimensions, heritage plays a key role in contributing to sustainability.

Today, the UN Sustainable Development Goals (SDGs) are somewhat synonymous with the concepts of 'global challenges' as 'any major trend, shock, or development that has the potential for serious global impacts' which require changes to the environments in which they operate [4]. Although the SDGs are intended to be 'universal', 'transformative' and 'people-centred' goals [5] they, in fact, focus to mitigate ongoing processes and practices that create or exacerbate crises rather than promote a transformational overhaul of existing structures and systems. On the one hand, it could be argued that SDG8 and SDG9—which promote economic growth, industry, infrastructure and innovations—stand at odds with sustainability depending on how nations interpret or enact sustainability measures. Ultimately, more production means more carbon emissions along with the increase in waste and environmental footprints, putting into question our ability to avoid tipping points that could lead to unsustainable living conditions on Earth [6]. On the other hand, innovations seen in technologies, particularly the 'interactions between the green and digital transitions' are at the forefront of political agendas, with their 'twinning' seen as key for achieving the UN SDGs [7] (p. 1). The expectation is that technologies will help decouple production from carbon emissions [8]. The recent foresight report *Twinning the green and digital transitions in the new geopolitical context* notes:

**Digital technologies could play a key role in achieving climate neutrality, reducing pollution, and restoring biodiversity.** By measuring and controlling inputs, and with increased automation, technologies like robotics and the internet of things could improve resource efficiency and strengthen the flexibility of systems and networks. Energy-efficient blockchain-based data management across the lifecycle and value chain of products and services could galvanise the progress towards a more circular economy and competitive sustainability. Digital technologies could also support monitoring, reporting and verification of greenhouse gas emissions for carbon pricing. Digital product passports enable enhanced material, component and end-to-end traceability and make data more accessible, which is essential for viable circular business models. Digital twins could facilitate innovation and the design of more sustainable processes, products, or buildings [7] (p. 2).

In that sense, the focus has been on fostering radical innovations to y decouple economic growth from their carbon footprint, or rather, the need for 'profound change of the economic and social policies' and the 'traditional view of economic progress' [7] (p. 10). However, radical innovations take time, and their successes are highly uncertain. They rely on the newness of ideas, and the readiness to adopt them, without much consideration of technological obsolescence and the life cycle of those technologies. The social dimension of sustainability is also many times absent in the analysis, including the need for either upskilling or reskilling of the workforce, and the impacts of the implementation of climate change response measures.

Taking this discussion to the urban level, in this article we discuss the opportunities and barriers of adopting eco-friendly ideas and technologies, and how adaptive reuse is key for urban spaces in their pathway to decarbonisation. We explore conceptual understandings of the economics of innovation, and how this might relate to the use or reuse of historic elements. We do this by the analysis of an extensive literature review to better understand the state of knowledge on adaptive reuse, and its relationship to innovation studies. We also critically assess and combine different strands of the literature on evolutionary economics, environmental economics, and system-wide cultural districts as cultural-led development models. The selection of literature used is predominantly based on secondary literature, using current peer-reviewed literature and/or publications from heritage organisations, although reflection on our own research has also been embedded into discussions. We provide historic mills as a case study to explore adaptive reuse using the valorisation and memorialisation of carbon-dependent legacies to think about mitigation from a just transition perspective.

It is increasingly recognised that 'cities are at the frontline of climate change' [9] and that they are not only 'hotspots' of climate impact but also essential for addressing mitigation, adaptation and resilience [10,11]. With a particular focus on advocating circular economies, it is essential to recognise the existing built environment's embodied carbon as relevant to carbon accounting. Through this lens, historic buildings can contribute to sustainable growth, promoting innovations that rely on what already exists, whilst acknowledging the role the historic environment plays in placemaking and enhancing a sense of place [12,13].

Finding new functionalities in heritage constructions can be seen as a type of innovation that simultaneously contributes to green growth and sustainable urban development. We also touch on the trade-offs and implications of what it might mean to ensure that heritage—widely defined- remains relevant in sustainability thinking, including existing examples of the heritage sector's adaptability and flexibility to respond to resource efficiency requirements and compete with a booming urban construction sector.

We explore the state of planning and its associated industries (construction and archaeology), the relevance of architectural innovations (as a type of innovation) in addressing sustainable pathways, and what some of the opportunities and barriers might look like from an economic approach. This angle is important for the sector to consider when advo-

cating for the continued use of historic buildings, and their potential to contribute to green growth and sustainable development.

## 2. Global Prioritisations of Sustainability and Urbanisation

Not long after the adoption of the 2030 Agenda, the New Urban Agenda (NUA),also known as Habitat III, was adopted in Ecuador. The NUA acknowledged the need to address the pressing issue of sustainability and urbanisation, flagging that 'if well-planned and well—managed, urbanization can be a powerful tool for sustainable development' and 'the source of solutions' [14] (p. iv). It is clear that 'urban areas hold more than half the world's population and most of its built assets and economic activities [15], thus we should not only anticipate the expansion and development of new urban spaces as changes to land use transforms regions [16], but also anticipate further concentration of climate risk in these areas. Goal 11 of the SDG, with the focus to 'make cities inclusive, safe, resilient and sustainable', further acknowledges the challenges ahead in terms of balancing these areas as centers of economic powerhouses, inequality and inequity, and extractive economies. Such resolve is dependent on urban planning and how to plan for growth sustainably, at least in the short-term ahead of a 'less is more' paradigm shift.

The New Urban Agenda loosely sets out 'standards and principles for the planning, construction, development, management, and improvement of urban areas along its five main pillars of implementation: national urban policies, urban legislation and regulations, urban planning and design, local economy and municipal finance, and local implementation' [14] (p. i). While commendable in spirit and intention, it falls short of establishing achievable goals and targets in which to enact its content. It does well to recognise the relevance of culture and cultural diversity in the pursuit of development initiatives, from protecting traditional practices to promoting retrofitting of urban areas and preservation of cultural heritage. Yet, although the action-oriented document's provision of a holistic and transformative approach to urbanisation centers around participatory and inclusive approaches, it overlooks the role of heritage as a tool towards participation, inclusivity, sustainable thinking and a circular economy.

Authorities and groups across the world continue to put emphasis on many of the highlights outlined in the NUA, such as innovation, good governance and planning. The World Economic Forum's *Industry Agenda Inspiring Future Cities & Urban Services*, for example, ranked environmental management and economic development as top priorities for urban sectors, and the most common challenge faced worldwide [17] (p. 10). Culture, on the other hand, has been hugely neglected and wrongly associated with less prioritised sectors such as leisure, public space and tourism (Figure 1). It is thus not seen nor understood as an untapped resource for transformation which is critical for tackling urban challenges. Conversations surrounding a city's heritage are reduced to preservation needs, acknowledged only for its offer of unique characters that can be monetised. The connection with sustainability remains largely unseen, although recent developments have seen heritage for the first time embedded into documents, most notably the COP27 Sharm el–Sheikh Implementation Plan [18].

Many heritage practitioners and scholars are actively addressing this knowledge gap [11] and the need to work across knowledge systems, industries and sectors. Certainly, a 'conscious shift' is emerging, with the need to adopt 'alternative indicators for development' that align social and environmental priorities alongside economic ones in which culture and heritage are seen as 'fundamental' in this process [19]. Much of the legwork involves understanding culture and heritage as more than material culture. Indeed, culture and heritage include knowledge systems, skills and practices of communities that have long dealt with localised environmental impacts and thus can identify local needs, and contribute to building awareness and capacity towards sustainable ways of living.

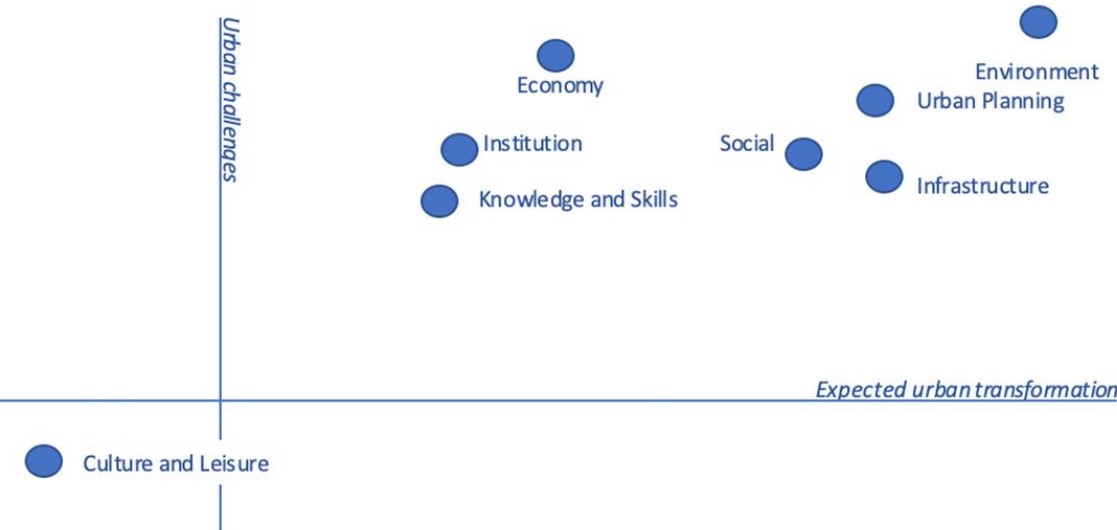

**Figure 1.** Source: World Economic Forum, Shaping the Future of Urban Development & Services Initiative. April 2016, Global Survey on Urban Services (October–December 2015).

As such, heritage can promote social cohesion and can help 'deliver positive environmental, wellbeing and regeneration outcomes' [20]. It has long been acknowledged that the social and cultural component of a community is key to more 'vibrant, harmonious and inclusive communities' enabling a sense of belonging and tolerance [21]. In particular, 'heritage has an increasingly important role in supporting sustainable growth [and] is a huge resource which can stimulate regeneration and growth in towns, cities and rural areas' [22]. Such heritage can also be a source of inspiration for social change and climate action [23]. Heritage, as tangible and intangible, also provides the stories and evidence of change over time—whether that be evidence of trauma, losses and damages, or that of adaptation and resilience—including climate impacts on the integrity and conservation of natural and cultural heritage [24]. In fact, heritage as an asset and management practice has long been adaptive to exogenous forces, including ongoing natural and human processes.

To that end, with heritage being an abundant and accessible resource that enables the enhancement of social, environmental and economic sustainability, it should be better understood and embedded into sustainability and/or climate adaptation plans moving forward [11]. All these opportunities across culture and heritage sectors provide the tools and instruments to meet environmental priorities such as mitigation and adaptation [11,23].

## 3. Adaptive Reuse and Its Relevance to the Construction Sector

In 2019, the *Global Status Report for Buildings and Construction* highlighted that decarbonising the construction sector is 'critical to achieving the Paris Agreement commitment and the United Nations Sustainable Development Goals' and that taking such climate action within the sector would be 'among the most cost-effective' [25]. The report also highlighted that the pace and scale of such a transition so far are not in line with climate targets needed to align with global commitments for a 1.5 °C limit. The world together will need to cut emissions by 45% below 2010 levels by 2030, meaning that in the next 7 years, we need to halve carbon emissions [25]. That translates into a cut in emissions by 8% each year, as opposed to what we now are experiencing with emissions reaching record highs [26,27]. 'Despite the clear warning on the extreme dangers of exceeding 1.5 °C warming from the IPCC, progress on new, more ambitious 2030 climate targets and participation in sectoral initiatives have stalled since COP26 in Glasgow' [28]. More needs to be done to decarbonise the building industry.

The construction sector, in which 74% of archaeology employment in the UK sits, accounts for over 40% of $CO_2$ emissions [26,29]. The Construction Leadership Council in the UK is focusing on regulatory, policy and technical frameworks to deliver a zero-

carbon industry, which will offer valuable performance framework metrics to support a reduction in resource waste on construction sites or in relation to logistics. But it is equally important for the heritage sector to recognise its role in championing developments and infrastructure investments for climate change mitigation, adaptation and resilience within the construction and its own sector. This has already begun: both internationally and nationally, the heritage sector (including archaeology) has started to address ways in which it can respond meaningfully to (a) understanding its wider role in contributing to the climate emergency, and; (b) recognise heritage as a powerful resource to climate action.

One such way is through adaptive reuse advocacy. England has one of the oldest building stocks in Europe with a fifth of all homes over a century old: 21% of all homes, 48% of all retail stock and 33% of all offices are over a century old, indicating the longevity and durability of existing stock's life cycle and the needlessness of demolition, wasting new materials, and increasing associated pollutants [30,31]. The built environment, in particular, needs to continue to think and practice innovatively so it can contribute significantly toward the decarbonisation path. To do so, circular economy thinking must be part of these conversations, particularly to enhance resource efficiency. Yet this is a complex and cross-cutting issue for many industries and sectors, with many potentially conflicting interests. Those involved in planning, land use and development weigh many factors in decision-making. These include, but are not limited to, questions such as: what should be saved and why? What are the incentives for demolishing existing building stock to make way for a new build? How should the displacement of communities be addressed in regeneration projects? To what extent should traditional skills be supported? Are new builds such as Passivhaus design standards more sustainable than retrofitting existing building stock? To what extent should financial viability for developers play a role in decision-making?

Currently, in the UK, there is an opportunity to improve circularity across specific sectors such as building and housing: for example, VAT on the maintenance and repair of the old building stock, but not for new builds, has been a huge issue of contention, seen by heritage sector advocates as a perverse incentive in favour of demolishing. Indeed, there have been cases where viability assessment models have been 'used to mitigate planning obligations in place to conserve and enhance the historic environment' [32]. Yet, demolishing existing buildings to replace them with new builds ignores studies that recognise the importance of existing buildings in actualising Net Zero targets by 2050. These studies include life cycle assessments' provision of a more complete measurement method for all carbon emissions (embodied and operational), and improved understandings of the whole building performance approach which demonstrates the inadequacy of energy performance certificate (EPC) assessments for historic and traditional buildings.

Adaptive reuse as the process involved in reusing and repurposing existing building stock, is seen as a green alternative to demolition by extending the lifecycle of the building, thus promoting a circular economy model. There are different theoretical approaches to adaptive reuse, which have coexisted in the last 50 years. Below (Figure 2) outlines some of these approaches [33] (pp. 4–5):

**Typological Approach**

Examines compatible uses for specific building typologies

**Architectural Approach**

Analyses the morphological relationships between old and new and the different design strategies (for example, the addition of new volumes inside, above, around the existing building)

**Technical Approach**

Focuses on building adaptations required to meet the needs of safety, comfort and usability (fire resistance, thermal behaviour, acoustic performance, etc.)

**Programmatic Approach**

Starts from the choice of a specific function and compares it with buildings available for reuse, in order to select a building suitable to accommodate such function

**Interior Design Approach**

Focuses on the "soft" values of the building (immaterial aspects, atmosphere, narrative), with the focus on protecting the "meaning" of the building rather than preserving its physical integrity

**Figure 2.** Theoretical approaches as understood by CITE. Please note the text is an exact extract from the publication.

Yet challenges endure for adaptive reuse as a dominant option in development. These include costs (including construction, marketing and maintenance), building condition and suitability, aesthetics and design, historic significance, environmental and social benefits/costs, carbon/GHG reduction, regulatory frameworks, changing demands of local communities, and cultural perceptions (including preference of new over old). Many of the above considerations align with recent considerations of adaptive heritage as an approach that 'recognises changing conditions and values' and the need to '(1) treat sites as living heritage, (2) employ innovative governance, (3) embrace transparency and accountability, (4) invest in monitoring and evaluation, and (5) manage adaptively' [34]. For example, Historic England, the advisory body to the Westminster Government, practises innovative governance in some capacity by adopting the broad term 'Constructive Conservation' to help manage change by providing a 'positive, well-informed and collaborative approach to conservation', which is flexible so that people can understand the relevance of the historic environment [35,36]. In this sense, the approach moves beyond the historic building as having inherent value and expands the idea of significance through collaboration with local communities, and its contribution to placemaking. Still, tensions continue between the preservation of historic buildings, out-of-date planning policies, and the urgency to better support adaptive reuse practices. Adaptive reuse, after all, on one level focuses on heritage material through repair, maintenance and retrofitting; but it also offers a paradigm shift for conservation processes and practices by prioritising societal and environmental demands over the maintenance of material, historic and design integrity only.

## 4. Types of Innovation

Innovation is not invention but rather refers to the translation of new ideas into real outcomes with commercial value [37] grounded in research, learning and creative thinking processes [38–40]. The push is towards meaningful combinations of different pieces of knowledge or collaborative efforts between knowledge systems [41]. In the *Oslo Manual 2018: Guidelines for Collecting, Reporting and Using Data on Innovation*, innovation includes new products, processes, marketing or organisational characteristics [42].

Moving forward, we suggest adaptive reuse of historic buildings is a type of innovation that bridges existing infrastructure with new opportunities. As such, it contributes to the transition to decarbonisation, responds to social demands, and leverages path dependencies

and cultural values that take time to change. But, as an innovation, there are challenges. Only through understanding the economics of innovation is it clear that adaptive reuse, as one type of 'architectural innovation', addresses the time gap of radical innovations, accelerating the transition to a green economy.

Innovations have been classified into different typologies, which consider the degree of newness of the innovation or how they impact end-users. For instance, the Propulsion Model classifies innovations according to the types of creativity the contribution requires and displays, and its impact, or the level to which it 'propels' existing ideas forward in the field [43]. Meanwhile, innovation experts see it as something that should 'be about more than products' and encompass new ways of doing [44,45], such as 'new ways of doing business or making money, new systems of products and services, and even new interactions and forms of engagement' between organisation and user.

Whilst there are many categorisations of innovation, below we use a classification that separates innovation into four types—incremental, architectural, disruptive and radical—based on the originality of the idea in relation to the presence of a market (Figure 3).

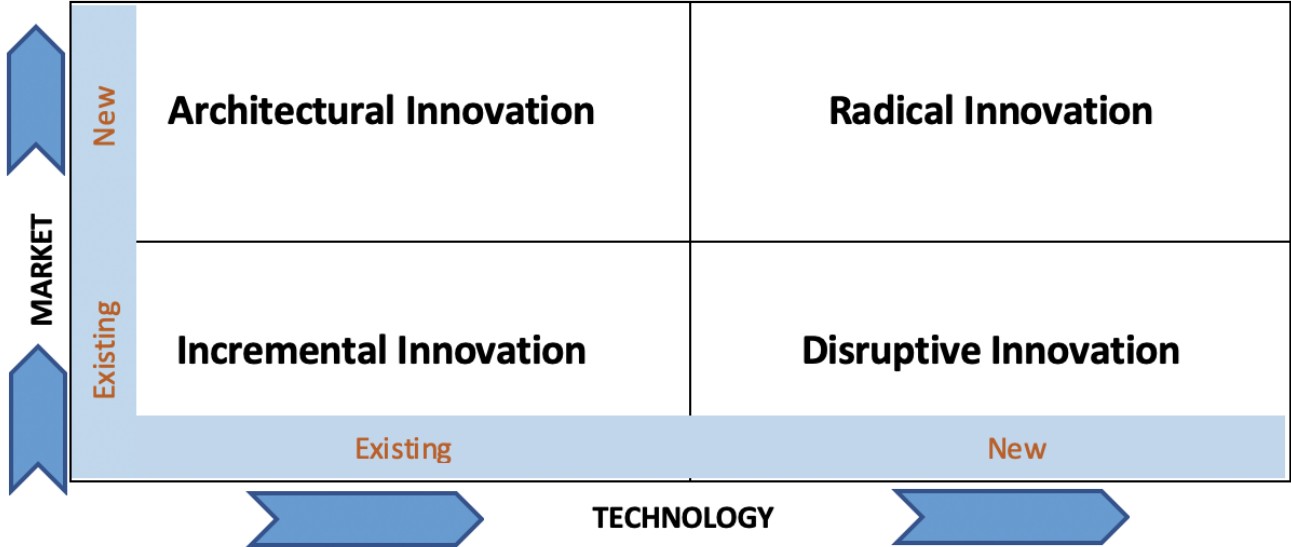

**Figure 3.** Lopez, 2015. Types of Innovation.

Incremental innovations refer to enhancements in products, processes, design or marketing [46]. They target an existing market. Disruptive innovations imply a reconceptualisation of technology within an existing market, while radical innovations mean both a reconceptualisation of the technology and the creation of a new market. Architectural innovations use existing components, but reconfigure them into new systems aimed at new markets. The components do not change, but the relationships between them do [46,47].

New technologies, products or processes that contribute towards social or environmental sustainable development and improved performances are eco-innovations. Eco-innovations include 'organisational and social changes for improving competitiveness and sustainability and its social, economic and environmental pillars' [48], thus are seen as fundamental for the transition to a green economy.

*Constraints in Adopting Innovation*

In general, stable structures include and involve strong ideological, social, financial, political or technological investments, that result in lock-ins and path dependencies that are hard to move away from. Thus, exercising adaptive reflexive thinking and practices are often complex operations. This is the same logic that makes the adaptive heritage concept so difficult to explore in practice. Shifting practices include hurdles such as economies of scale, learning, costs, path dependencies and superior cost-benefit analyses considerations [49,50]. Often, economies might actually be 'locked-in to inferior technology paths', with superior

alternatives not being able to compete or gain a footing due to these hurdles and considerations [51,52]. An obvious example here is the current carbon-reliant technologies in place, which have had huge investments, making alternative options less attractive. Alternative options, such as Electric Vehicles, require significant investments in infrastructure including energy supply, charging stations and other maintenance support. These sorts of long-term considerations and necessary investments influence the purchasing confidence and deployment opportunities of eco-innovation alternatives, and so take significant periods of time [53]. A potential lack of purchasing confidence makes it difficult for consumers to access such innovations, if at all, particularly with their high prices alongside an uncertainty of whether the investment is worth it or not. In a built heritage context where maintenance and repair costs are already considerable, these barriers decrease user buy-in.

In economic terms, there is a challenge, or the so—called *double externality* problem. 'By ( . . . ) spillovers, we mean that (1) firms can acquire information created by others without paying for that information in a market transaction, and (2) the creators (or current owners) of the information have no effective recourse, under prevailing laws, if other firms utilize information so acquired' [54] (p. 16). The spillover effect can occur during the research and development (R&D) and innovation phases, with developers unable to appropriate all benefits from their own activity [55]. This suggests that the social benefits of eco-innovations exceed the technology's market profit, leading to an under-investment in them because companies cannot adequately appropriate them [49,50]. Another externality occurs during the diffusion phase: because incentives of the market are distorted, incumbent technologies that are already settled in the market compete under equal conditions to new green technologies. Having already expanded and grown at a time when there were fewer concerns about sustainability [49,50] and benefited from economies of scale, they are then able to be competitively cheaper than greener alternatives [52].

Financially, eco-innovations tend to involve high costs in an uncertain environment: not only do innovations intrinsically carry success uncertainty and a high risk of failure, but there is also a lack of historical data to assess associated costs and benefits. This is at the heart of why capital markets fail to provide funds for eco-innovations [49,56,57]. In periods of competing technologies, this becomes even more problematic as there is additional uncertainty about which of them will succeed.

Despite the need for eco-innovations and the vast uncertainty surrounding its success, the financial system itself discourages innovation by focusing on short-term profits, and manipulation of stock value by companies investing in their own stock. These practices have had detrimental effects on innovation expenditure, highlighting that the system needs reform, as its current state simply is not structured to encourage radical innovations [49,57–59].

Added to the above barriers is also the need for institutions to be more adaptive and flexible to change so that they can easily and comfortably modify business models, companies' routines and competencies [52]. This idea of explaining how, why and at what rate new ideas and technology spread was explored by American professor of communications Everett Rogers in his theory of diffusion of innovations [60]. He argued that the spread of a new idea or technology is reliant on: what the innovation is, those that adopt it, how it is communicated more widely over time, and the social system into which it is embedded. Adopters, for example, are central in ensuring an innovation reaches a critical mass, and this process itself goes through five stages: knowledge/awareness (exposure without information), persuasion (actively seeks information), decision (to adopt or reject), implementation (judgement of usefulness), and confirmation/continuation (decision of continued use) [60]. Both time and coordination are key here—radical innovations, for example, are system-wide and collective, involving new synergies between often disconnected activities and between different actors across sectors and locations. This is where policy agendas and establishing frameworks, guidelines and policies are so crucial, yet take a long time to establish. Such policies would be needed to coordinate disconnected activities, actors and sectors and to shape radical innovations so that they have the potential to address societal priorities and challenges, such as tackling climate change [52,56,61,62].

Certainly, 'technological improvements in eco-efficiency alone will be insufficient to bring about a transition to sustainability' [63], not least due to myopic fixes and ideological and cultural dependencies on unsustainable practices, industries, and economic systems. The implications of adopting different innovation are seismic: radical innovation disrupts, replaces or destroys existing systems and processes in place, to then enable change processes and practices. The reality of restructuring involves complex considerations and the interplay between technological upgrades, scientific and ideological advances, institutional flexibility, and strategies across multiple parties amongst other variables (see Creative Destruction [64]). This is separate from incremental innovations that can simply relate to technical progress, and might not face the same constraints and challenges as the former. Meanwhile, architectural innovations do not require significant changes in businesses, technologies, organisations and institutions, but rather are built on existing capabilities, reducing levels of uncertainty. Architectural innovations provide a bridge between radical innovation and uncertainty, building on existing skills, knowledge and technology, but with a new approach or purpose [65]. Thus, there is an urgent need to rethink existing processes and practices, traditional practices of investment and profit, path-dependencies, and support risk-taking ventures for radical change and transformative shifts. This could help support new and needed adjustments to social, economic, and ecological systems in place to better align with emerging technologies [47].

## 5. Heritage, Adaptive Reuse and the Circular Economy

Adaptive reuse, as an example of architectural innovation, involves finding a new use for an existing structure different from its original purpose. Reusing, or repurposing, the existing building stock is considered highly sustainable even if just by acknowledging past resource expenditure in calculating resource and energy efficiency. It also provides continuity for the urban landscape to maintain its sense of place, character and identity, whilst also providing opportunities to transform it through renovation, restoration and redesign of disused, under-used or abandoned buildings. In some instances, it can be an opportunity to consider broader community needs and changing areas of local significance and value, which change through time. Of course, displacement of communities is also a consequence of regenerated historic urban landscapes as high prices push communities out, and with that their cultural practices that impact placemaking, which also should not be ignored (but will not be discussed in this paper).

Historic buildings offer an additional incentive to transform the traditional linear economy of take-make-dispose into that which is more circular, with the added value of historic buildings and heritage benefits providing a sense of place, connection to the past, unique local character and vibrancy [66]. Yet debate continues as to whether the reuse of historic buildings is more or less efficient than new builds. Currently, in England, a public enquiry into a proposal to demolish the Marble Arch Marks & Spencer's building dating to the 1920s has spurred questions on whether demolition can 'ever be better than retention' [67]. Meanwhile, proposals to the National Planning Policy Framework via Paragraph 161 aim 'to clarify that significant weight should be given to the importance of energy efficiency through adaptation of buildings, whilst ensuring that local amenity and heritage continues to be protected' [68]. Indeed, there exist life cycle assessment comparisons between renovation of existing buildings versus new builds in which retrofit houses outperform new builds in both the assembly and operational stages whilst new builds perform better at the end-of-life stage (subject to standards) [69]. However, others demonstrate that, in fact, this is a complex comparison which requires a selection of appropriate boundaries and comparative indicators not necessarily included in current Life Cycle Assessments used in England. With these considerations, the renovation has been 'found to help reduce environmental impacts associated with the lifecycle of building components by 53–75%' [69–71].

Understanding embodied carbon in the construction and building sector should not be ignored during carbon accounting. This carbon relates to that emitted for the

extraction, manufacturing and transport of materials used in the construction of a building. In the case of a new build, embodied carbon accounts for 35% of the carbon emitted in a 50-year period of analysis [72,73]. Meanwhile, the embodied energy for renovation is approximately 7% over the lifetime of a building; in the case of repair and maintenance work, it represents less than 1%. Renovating historic buildings is far better than demolishing them in terms of carbon emissions and resource efficiency: embodied carbon will be lower and the operational energy (the carbon consumed while using the building) has the potential to equal that of a new build. Refurbished buildings imply similar levels of energy consumption, with lower use of new materials and water in comparison with new buildings [73].

Thus, the very nature of already existing, historic and traditionally constructed buildings outweighs the environmental cost of a new build, at the very least during the assembly, operational, and (if required) demolition stages of the process. The demolition and subsequent replacement of historic buildings create new embodied carbon—the indirect carbon consumed during the construction/repair/retrofitting process- that can and should be avoided [74]. Also, historic and traditionally constructed buildings tend to avoid carbon-intensive materials such as concrete and steel and are generally constructed with solid walls of masonry or earth, or timber frames with solid infills, and use porous material which allows them to heat up or cool down slower and better tolerate moisture exchange [75,76]. Despite historic and traditionally constructed building materials being resource efficient, they continue to be 'stigmatised as "hard to treat", or energy-hungry, despite the evidence of several thousand years of proven effectiveness in a low-carbon, low-energy environment' [77].

Further, evidence suggests that reusing historic and traditionally constructed buildings benefits people's well-being [78], and demonstrates that 'interaction with heritage or the historic environment can be a positive factor in supporting individual and community well-being' (for evidence, projects, case studies and strategic documents, see Historic England's Wellbeing website [79]). Repurposing existing stock also contributes to placemaking in shaping a sense of place. Thus, the wider historic environment is increasingly understood as critical for new schemes driving placemaking and regeneration: to enhance the delivery of public value through urban design, they are increasingly incorporating the adaptive reuse of historic buildings. This includes approaches that repurpose traditional materials or which harness unique and distinctive histories when exploring how to attract investment and encourage cultural activity [80].

Heritage as a starting point for urban development and urban regeneration is recognised to have a strong role for users and local communities. 'Cultural heritage, especially if understood as a system and process [81], goes far beyond physical qualities and is embedded in the lives of many communities not only through the use of cultural heritage but also through sociological dimensions like heritage identification [82], a sense of feeling at home [83,84] and values from the past that contributes to these feelings' [85]. From three European Projects (HerO (Heritage as Opportunity), Hall and Model project and COMUS (Community-based Urban Development) [86]), a five-pillar model (Figure 4) was developed as a means to consider heritage as a starting point for urban regeneration [85]:

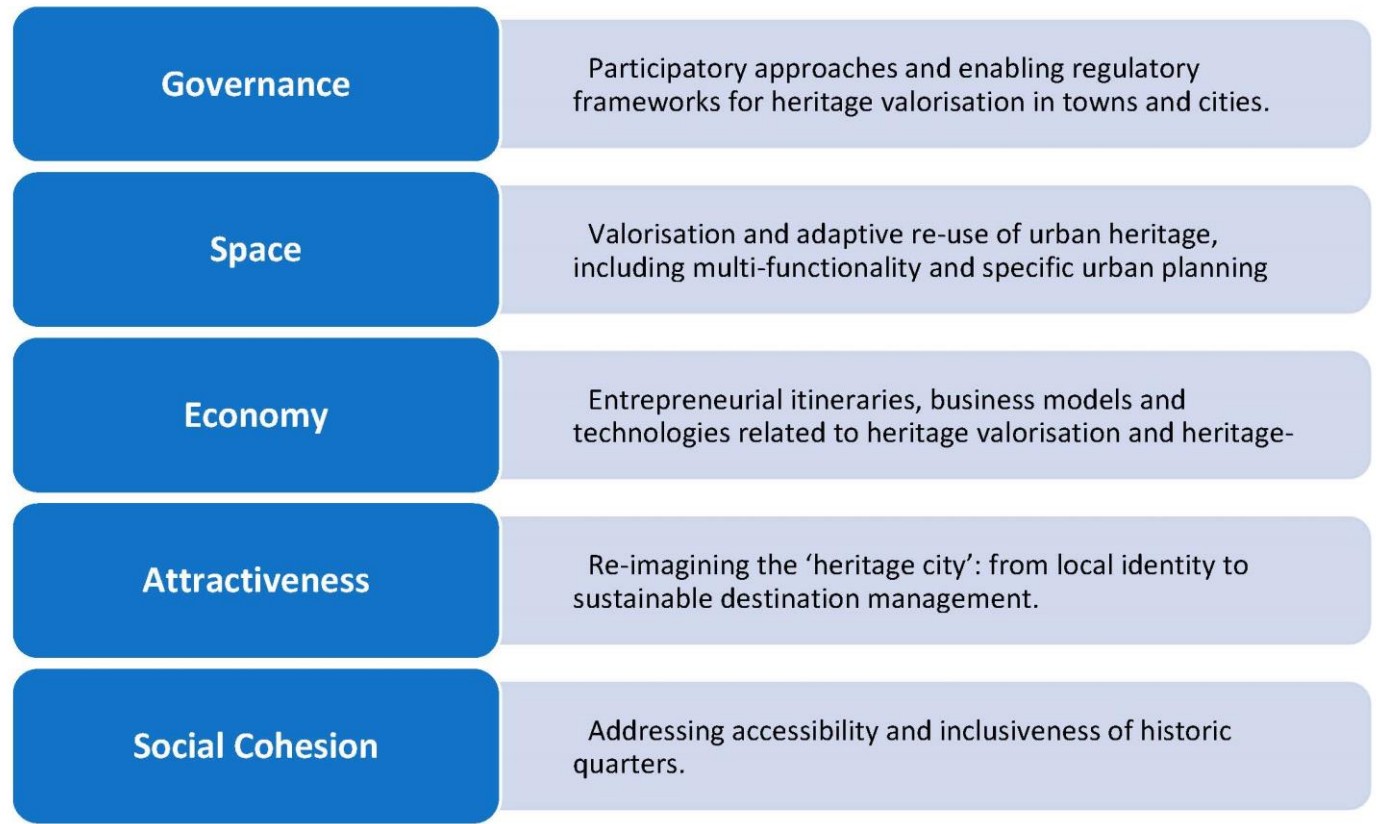

**Figure 4.** Heritage and Urban Regeneration five-pillar model.

The valorisation of local heritage—e.g., cultural, archaeological, biological, agricultural, artisanal and environmental—is best done through an integrated approach in which heritage is understood as part of a whole urban landscape rather than seen through a piecemeal approach. This includes understanding the benefits of heritage adaptive reuse through an interdisciplinary lens, whereby wider socio-economic outcomes—from climate resilience to job creation—become evident. It also includes thinking flexibly about heritage orthodoxy and what the sector understands as favourable processes for the perpetuity of heritage. Architectural innovations, for instance, in the historic environment can contribute to green growth by creating new job opportunities and economic activity, while reducing the carbon footprint. Current estimates suggest that for every €1m invested in energy renovation work of existing building stock, 12 to 17 new jobs are created [87,88]. Adaptive reuse involves refurbishment activities, which creates new jobs that require either traditional and/or new skills [73]. Yet, heritage skills in the construction industry are struggling to survive, despite being vital if we are to maintain and repair existing buildings to ensure the targets of the Paris Agreement are met. Such skills include 'essential crafts such as bricklaying, carpentry and joinery, painting and decorating, plastering, roofing and stonemasonry' to name only a few [89]. It is suggested that the adaptive reuse process in itself creates more employment than that which a new construction would [90].

The repair and reuse concept of urban landscapes offers more than just responding to a single concern. Historic parts of cities are known to attract more tourists and creative talent, which can provide investments and potential economic regeneration for the area. It is a misconception that conserving urban heritage hampers growth and development, one that is increasingly being challenged [91]. Of course, it is again relevant to flag that the regeneration of historic areas comes with other serious considerations, such as the unethical displacement of communities that in their own right can sustain living heritage of areas. An example is the recent regeneration of London's King's Cross as an important landscape of industrial heritage with a significant number of heterogeneous communities. King's Cross

now promotes itself as carbon neutral, through the building of 50 new energy-efficient buildings and the restoration of 20 historic buildings, in addition to carbon offsetting 'historic emissions' (i.e., embodied carbon). Other intangible aspects of heritage, however, tend to be overlooked (e.g., street performances, community arts, cultural events, or other demonstrations of culture and heritage) which are equally important when considering adaptive reuse, because it is those characteristics that contribute towards creating that sense of place that people value. The 'benefits associated with historic buildings and places are often interrelated, with improvements to an area's image and sense of place helping to generate new economic activity and investment, which in turn can contribute towards enhancing the quality of life for all' [92] (p. 3). It is also clear that quality of life includes the ability to perform and access cultural practices, and the ability to shape places through this kind of heritage. It is often these costs and trade-offs that make climate mitigation complex: The neglect to recognise the wider benefits that heritage can provide to urban regeneration to move towards climate resilience is problematic. Heritage-led regeneration projects tend to enhance the inclusion of the voluntary and community sector, and provide space for such local community engagement, but this must not be assumed as a given and should be planned into regeneration through collaborative efforts with communities. For example, investment in London's East End historic streetscape Brick Lane led to the revitalisation of the community, with Brick Lane now the base for many Bengali festivals and other cultural activities (Figure 5). Again, such successes are not without their tensions, which include the popularisation of places leading to gentrification and concerns regarding the displacement of established communities.

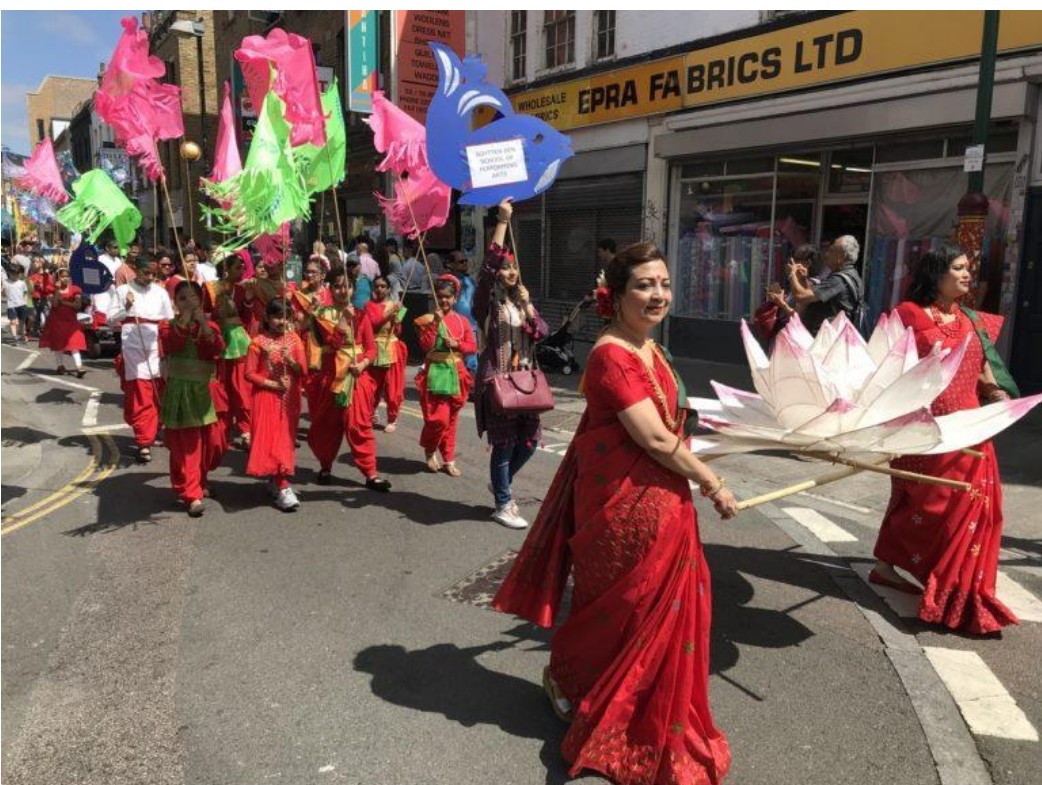

**Figure 5.** The Brick Lane Boishakhi Mela Grant Parade. Credit: Dave Stuart. instagram.com/ shoreditchstreetarttours (accessed on 1 October 2022).

Other examples of adaptive reuse in London include the conversion of Sir Giles Gilbert Scott's abandoned Bankside Power Station into the Tate Modern, one of the most iconic museums of the global city; or the regeneration project of the Battersea Power Station as a shopping and leisure destination. What these stories suggest is that there is more to adaptive reuse than merely resource efficiency. It is about a wider integrated approach to

thinking about social, economic and environmental sustainability through design, planning and community considerations.

More importantly, as the movement of people continues to shift towards urban areas and the resource implications that movement involves, it is timely to focus on future needs of urbanisation, or, emerging towns and cities prone to urbanisation. Practices of circularity must become mainstream if urbanscapes are to fulfil the obligations of the Paris Agreement. For these future scenarios, we explore what was once the powerhouses of the Industrial Revolution: historic mills.

*Mills of the North, Adaptive Reuse and Green Growth*

The Yorkshire and the Humber region, like the rest of England, is being impacted by climate change, and this is projected to continue for the foreseeable future. The region's wide range of landscapes including uplands and lowlands means that the nature of climate change will vary across the region. Meanwhile, the region sees itself as 'well-placed' to explore the future of industry with its 'strong historic base' [93], which is key for its move towards suitable climate change measures.

To briefly contextualise, Britain is considered home to the Industrial Revolution, which had profound impacts on modern society. As such, remnants of this time, whether intangible or tangible, record a fundamental period of world history and transformation. The North Powerhouse was home to the Industrial Revolution, remnants which remain a dominant feature of landscapes and communities. The Industrial Revolution—climate heritage in itself- was an era of transformation to the economy, society and environment that changed ways of life, places and landscapes across the world and continues to influence them today, not least through its legacy of extractive industries and economies, all heavily carbon-dependent—which has resulted in the current climate crisis. The remains of such industrial culture—or Industrial Heritage, include 'buildings and machinery, workshops, mills and factories, mines and sites for processing and refined, warehouses and stores, places where energy is generated, transmitted and used, transport and all its infrastructure, as well as places used for social activities related to the industry such as housing, religious worship and education' [94]. These facilities are rich with historical, technological, architectural, scientific, natural and socio-economic values.

While industrial sites include a range of physical remains that continue to characterise places and landscapes associated with industrialisation—such as railway, water and other transport systems, energy generation and distribution systems, production machinery and establishments, and agriculture—they also include stories. Our industrial past goes beyond physical assets, and can tell a story of innovation and the societal impacts of new technologies of the time, and more importantly, understand those impacts over a long period of time. Industrial heritage can help communities reconcile 'our past with our future' by celebrating 'a proud industrial past' and supporting regions to 'take this part of our identity forward without locking the region into congestion, polluted air and climate breakdown' [95].

During the recent industrial transition during the 1980s, communities felt ignored and abandoned. This legacy of being left behind is now being acknowledged in different parts of the North as the country moves towards a green economy: Yorkshire and Humber Climate Commission director Professor Andy Gouldson had previously commented that Yorkshire 'knows all too well what a brutal energy transition feels like from 30 years ago and the last thing we want to do is repeat any aspect of that' [96]. The Commission's work is based on fostering climate resilience and adaptation whilst also encouraging 'a just and inclusive transition and climate actions that leave no one and nowhere behind' [97]. Indeed, this aspect of 'adaptation is a necessary strategy at all scales to complement climate change mitigation efforts' [98] (p. 6), which includes thinking about how different aspects of understanding people's industrial past (from storytelling to its physical presence in neighbourhoods and landscapes) play a part in conversations about place-based climate action and long-term planning for community building, adaptive reuse, urban renewal

and cultural regeneration, and the transformation of redundant historic industrial sites (for more on this, see work from the Climate Heritage Network's 'Just Transition' focus).

Historic mills have a profound impact on the physical character of a landscape as well as its cultural and historical legacy. They are also defined by their environment, either by their power source or geographical location. Recent research highlights the 'scale of the opportunity' in repurposing recently refurbished and currently disused mills [99]. Indeed, the UK Government has flagged that to create a Northern Powerhouse, there is a need to 'celebrate the North's rich heritage, and use our past to build a brighter future' [100]. While historic mills are vulnerable to and dependent of external circumstances, they continue to shape the distinctive character of the towns or cities they reside in, and help define the identity of their community [99].

Meanwhile, there are some 688 historic mills currently vacant or under-utilised in Greater Manchester, Yorkshire and Pennine Lancashire alone [101]. Many others have been lost forever: for example, it is estimated that 45% of historic mills have been demolished in Greater Manchester in the last 30 years [101]. Yet these mills evidence and tell stories of the lives and livelihoods of men, women and children during a time when Britain was a world leader in industrialisation.

Just one of the many kinds of industrial heritage present across England which is a building of local interest is the Queen's Mill, located at the River Aire in central Castleford (Figure 6). The Mill is owned by the Castleford Heritage Trust (CHT), established in 2000 to 'promote the community's heritage and culture to build a strong, successful community' and to 'use natural as well as cultural heritage as a vehicle for regeneration and improving educational opportunities' [102]. The Trust also promotes access through community involvement to industrial heritage, and as of 2013, purchased and moved into the Queen's Mill, kicking off a major refurbishment project.

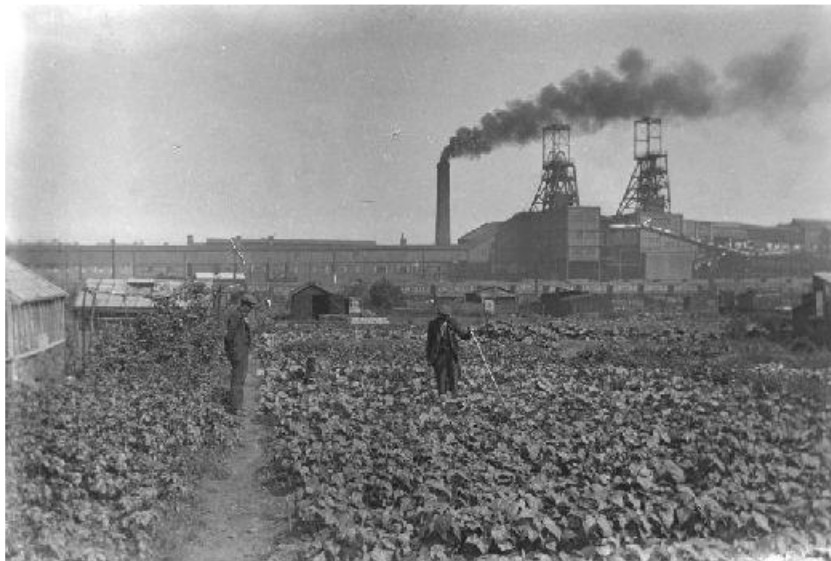

**Figure 6.** Fryston Colliery and Castleford allotments. Copyright: Wakefield Museums & Castles, Wakefield Council.

The Castleford Queen's Mill demonstrates how industrial heritage can enable an exhausted town in decline to be revived if included in wider efforts to regenerate towns. Through volunteers and crowdsource funding, CHT managed to turn the landmark into an attractive space and has also begun adding mitigation measures, e.g., solar photovoltaics to its rooftop. Community activities run from the mill, which also has an arts and exhibition space and refurbished rooms for businesses. The group also worked to get the site milling again to produce and sell flour and support more work [103].

Castleford has taken a proactive inclusive approach that helps to better understand how areas with rich heritage can explore the wider benefits of revitalising and repurposing historic assets. In relation to climate change mitigation and adaptation measures, such an approach not only addresses the need to reuse and repurpose existing (and under-used) building stock but also highlights an inclusive process towards local regeneration through rethinking, reorganising, and adapting to changes. From a social perspective, the story of the Queen's Mill, with its local rather than national significance, is part of a wider industrial heritage story and community story. It gives a nod to the human story of the many consequences of industrial transitions, the potential of community action and the role the past has in thinking about change.

Castleford is just one small example of the potential for adaptive reuse. Repurposing historic mills contribute to a low-carbon and resource-efficient economy, while also pushing reflexive thinking and adaptability in how the heritage sector responds to change.

## 6. Final Thoughts

Adaptive reuse is essential for a circular economy. It is an extension of a value system existent in daily lives, illustrated by the proverb 'waste not, want not', first recorded in 1576. While the proverb's intent implies the need to use one's resources wisely to avoid poverty, it echoes the concepts of resource efficiency through maintenance, repair and reuse. The practice of a circular economy has long been embedded in daily practices, albeit somewhat overshadowed by the current Disposable Society challenge.

This paper explores adaptive reuse from an economics of innovation perspective, and further shows how adaptive reuse can contribute to larger sustainability, beyond the architectural sector. It does so to highlight the importance of adaptive reuse for urban areas if they are to meaningful address and meet Paris Agreement targets. The adaptive reuse of historic buildings involves combining existing ideas in new ways, resulting in real outcomes with commercial value, and can be complementary to disruptive or radical innovations. Promoting adaptive reuse across policy and frameworks, and ensuring it is supported by financial incentives, is a means to address mitigation and adaptation with both immediate and long-term action. Since radical and disruptive innovations require overcoming technological, financial, economic and institutional changes, architectural innovations can support quick sustainability transitions and avoid barriers (e.g., path dependencies) seen across urban spaces.

In our article, we propose that from a life cycle approach inclusive of both embodied and operational carbon, the adaptive reuse of buildings contributes not only to lower costs, resource efficiency and reduced carbon emissions but to social and economic benefits that help create the social fabric in land use. The integration and creative use of heritage fosters a sense of belonging and social cohesion that can positively impact community wellbeing, including life satisfaction. The revitalisation of heritage involves supporting local knowledge, skills and practices and inclusive, participatory activities. It broadens the audience by finding new uses for buildings that would otherwise be vacant or lost forever. It also creates new ways for people to engage with heritage. The adaptive reuse of historic buildings drives value creation to society by contributing to green growth, while forging a sense of belonging and attachment to a place recognising the changing needs of local communities. Architectural innovations in the built environment enable the repurposing of historic buildings to meet new societal demands and help the sector rethink real and assumed barriers related to heritage asset renovation. Complementing NUA and economics of innovation approaches, heritage can be seen as a driver of urban sustainable development processes, contributing to future generations' well-being by promoting cultural-led regeneration processes that make use of already existing embodied carbon, while considering social inclusion.

Focusing on a literature review across heritage studies, economics and sustainability, our analysis and examples contribute to the decisions surrounding adaptive reuse and its role in sustainable urban development. The reuse of existing building stock is essential if

nations are to realistically move towards Paris Agreement targets. However, the uptake of adaptive reuse is low, with perceived barriers and restrictions challenging the benefits of re-purposed buildings. Yet, not only does this solution offer a credible way towards reducing GHG emissions, but it also offers a wide range of societal benefits, or reconciliations with past legacies, that are vital for progress within social sustainability. Both historic buildings and traditionally constructed buildings survive tumultuous external drivers while creating a sense of belonging and attachment to places, creating social bonds and forging social cohesion, and maintaining elements that define their identity and the identity of a place, which are all drivers of social stability [12,104,105]. Rethinking how we value heritage in addition to how we acknowledge and reinvigorate the principles of reuse, repair and recycle—and of course, maintain finished products that remain functional, durable, and made of quality material—is essential if we are to address the issue of sustainable development.

This area of study can benefit from further research, such as the quantitative analyses of heritage adaptive reuse contributions to social, environmental and economic sustainability, particularly using macro-indicators and assessing target achievements. Multi-criteria analysis could be also be applied to prioritise between macro-indicators [106]. Additional work on developing a vulnerability index that identifies geographical areas that would particularly benefit from regeneration processes of historic buildings would be beneficial [107], including a better understanding of adaptive reuse and regeneration processes, community displacement and values of heritage.

**Author Contributions:** Writing—original draft, H.M. and B.D.D. All authors have read and agreed to the published version of the manuscript.

**Funding:** This research received no external funding.

**Institutional Review Board Statement:** Not applicable.

**Informed Consent Statement:** Not applicable.

**Data Availability Statement:** Not applicable.

**Conflicts of Interest:** Authors declare no conflict of interest.

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
