# Peer review of "Adaptive Thinking in Cities: Urban Continuity within Built Environments"

_climate, doi:10.3390/cli11030054_

Round 1

Reviewer 1 Report

CONTENTS

The paper is very interesting and well written, although it is a little unbalanced, giving too much attention to social and economic issues and overlooking a specific focus on architecture.  As a solution, section 9 could be implemeted with more technical comments and information about the adaptive reuse intervention of mills and hopefully extended to other examples. Section 5 and 6 could be merged giving and wider idea of what is adaptive reuse.

LANGUAGE AND STYLE

Please chck spelling in the title of section 8, check references which are missing in the text.

Section 3 could be eliminated and turned to figure.

Check figures and figure numbers, the last one is missing.

Author Response

Comments received from the reviewer are outlined below, with the author responses in bold.

  1. The paper is very interesting and well written, although it is a little unbalanced, giving too much attention to social and economic issues and overlooking a specific focus on architecture.  As a solution, section 9 could be implemented with more technical comments and information about the adaptive reuse intervention of mills and hopefully extended to other examples. 
    • We have revised the manuscript so that the focus is of sustainability more holistically, moving away from the architectural focus. While we have not added the more technical comments in section 9, we have framed it as a way to think about the relationship between carbon-reliant economies and just transition.
  2. Section 5 and 6 could be merged giving and wider idea of what is adaptive reuse.
    • Some sections have now been deleted and/or merged
  3. LANGUAGE AND STYLE
  4. Please check spelling in the title of section 8, check references which are missing in the text.
    • Done. Thanks!
  5. Section 3 could be eliminated and turned to figure.
    • Not sure if this relates to the construction section but all sections have been revised by deletion or merging
  6. Check figures and figure numbers, the last one is missing.
    • Checked; with some images deleted.

The authors would like to that the reviewer for their thoughtful feedback to the manuscript. We have taken the time to revise the entire manuscript quite significantly, and hope that it now comes across clearly.

Reviewer 2 Report

Overall, I quite like the topic and believe it is relevant. Nevertheless, the paper has a few shortcomings that could and should be solved before publication.

The introduction is a bit “all over the place”, it refers to a few rather old landmarks with the Club of Rome and even Humboldt, where the link to the paper topic isn’t entirely clear. The SDGs are introduced, but the NUA isn’t, though the abstract refers to it. Furthermore, progress on SDGs and changing understanding towards its transformative notion isn’t considered at all, neither do authors refer to the “heritage” aspects in global agendas. Here I suggest to link the introduction more to the actual topic of the paper.

The paper is framed along sustainability as a concept. While the concept itself is explained briefly, the application and understanding in the paper is a bit blurry. Authors assess the different values of heritage, including social and for CC adaptation/mitigation (though this is rather implicit), but they don’t really distinguish/assess this by sustainability dimensions. Either sustainability should then be removed/not put that strongly as framework or the analysis should be structured more along the sustainability dimensions. Given the paper is submitted to “Climate” this component should be fleshed out more strongly in the introduction and the whole rationale/paper.

The whole paper is written in a very descriptive way, it could benefit strongly from shortening the text. Furthermore, there is no method chapter, at least this should be added briefly what kind of analysis based on which kind of data/literature is carried out. I am also lacking a reflection on limitations.

The final thoughts chapter states “This paper explores how adaptive reuse of heritage is a type of architectural innovation.“ I would believe the paper promises more, rather showing how adaptive reuse can contribute to larger sustainability, beyond the architectural sector. But this is exactly what is not entirely clear throughout the paper and what I believe should be strengthened, as the thought that adaptive reuse is useful in multiple ways is a very valid one.

94ff: suggest to add focus of the other agendas listed as not every reader might be familiar with it, or to skip, as it might not be needed to give all?

128: reference missing

181ff: sentence unclear, do you refer to UK or global scale?

Chapter 4: unclear what role archeology plays for achieving the 1.5 degrees goal, either cut or elaborate

Figure 3: please revise, the COP is not a framework, neither are IPCC reports

208: add reference

212: add reference

Chapter 5: I am missing a reflection on the suitability of historic buildings/fabrics under CCA aspects, e.g. when it comes to non-sufficient insulation etc, and retrofitting demands vs conservation of historic fabric

Figure 4 (p 7) is not elaborated further in text, there is no reference/further usage of these different approaches, hence my suggestion is to either add or remove figure

There are 4 figures 4

Chapter 6 introduces different forms of innovation, however again, after that description the types aren’t further used, raising questions why they have been introduced, likewise chapter 7 hardly touches on the exact paper topic, but is rather again another context

388/437/458/506: reference missing

Chapter 8 is kind of the centerpiece, where finally the paper topic is being discussed, but it mixes identified with hypothetical potentials, which could be distinguished a bit more

Chapter 8 already gives examples, but then chapter 09 focusses on only one example, my recommendation would be to cluster the examples in one chapter

Others:

There are quite some typos in the paper & figures, hence proofreading is recommended

With respect to figures the paper copies several ones from other publications, I would recommend to reconsider if they are all needed, e.g. the sense of figure 2 i and its relevance for the point the authors are making isn’t entirely clear to me

Author Response

Comments received from the reviewer are outlined below, with the author responses in bold.

  1. The introduction is a bit “all over the place”, it refers to a few rather old landmarks with the Club of Rome and even Humboldt, where the link to the paper topic isn’t entirely clear. 
    • The intro has been revised, including deletion of the Club of Rome, etc history.

  1. The SDGs are introduced, but the NUA isn’t, though the abstract refers to it. 
    • The explicit link to the NUA has been slimmed down, with more reference to discussions happening on an urban level, to make it more focused.

  1. Furthermore, progress on SDGs and changing understanding towards its transformative notion isn’t considered at all, neither do authors refer to the “heritage” aspects in global agendas. Here I suggest to link the introduction more to the actual topic of the paper.
    • This has been deleted, and rather the introduction is more targeted on urban challenges.

  1. The paper is framed along sustainability as a concept. While the concept itself is explained briefly, the application and understanding in the paper is a bit blurry. Authors assess the different values of heritage, including social and for CC adaptation/mitigation (though this is rather implicit), but they don’t really distinguish/assess this by sustainability dimensions. Either sustainability should then be removed/not put that strongly as framework or the analysis should be structured more along the sustainability dimensions. 
    • See above

  1. Given the paper is submitted to “Climate” this component should be fleshed out more strongly in the introduction and the whole rationale/paper. 
    • The sustainability concept is more developed before section 4 and conclusion addresses in a more detailed way the connection between adaptive reuse with sustainability in cities.

  1. The whole paper is written in a very descriptive way, it could benefit strongly from shortening the text.
    • Some examples in the last section were removed and some graphs/pictures were deleted to shorten the paper.

  1. Furthermore, there is no method chapter, at least this should be added briefly what kind of analysis based on which kind of data/literature is carried out. 
    • A brief few sentences are now added to the methodology of the paper. See end of page 2

  1. I am also lacking a reflection on limitations. 
    • In the conclusions we added a paragraph on further research addressing the limitations of the article.

  1. The final thoughts chapter states “This paper explores how adaptive reuse of heritage is a type of architectural innovation.“ I would believe the paper promises more, rather showing how adaptive reuse can contribute to larger sustainability, beyond the architectural sector. But this is exactly what is not entirely clear throughout the paper and what I believe should be strengthened, as the thought that adaptive reuse is useful in multiple ways is a very valid one. 
    • The conclusion has been revised and now suggested its wider contribution beyond architecture.

The authors would like to that the reviewer for their thoughtful feedback to the manuscript. We have taken the time to revise the entire manuscript quite significantly, and hope that it now comes across clearly.

Reviewer 3 Report

1)

Lines 415-420: Renovating historic buildings is much better than demolishing them...

The question concerning the renovation or demolition of historic or protected buildings is a very serious one. It is not determined exclusively by a factor of cost-effectiveness, which may be secondary in this field. There are many legislative frameworks, such as Italy's, that categorically prohibit the demolition of such existing heritage.

Please clarify this approach !

2)

The article suggests some good perspectives, suggesting some macro-indicators, but it is developed in a purely theoretical form. 

3)

No connection is made between the approaches in Figure 4 , the LCS approach in Figure 5, as well as the five pillars shown in Figure 8. Nor are relative weights and/or incidences of these parameters determined.

4)

No clear methodology is defined, nor a model to be tested against one or more case studies (the latter are only described in a purely theoretical way).

5)

The bibliography is extensive and relevant, which suggests a thorough analytical phase. But the latter does not translate into concrete methodological elaboration.

6)

The following references are recommended:

DOI: 10.3390/SU12166432

DOI: 10.3280/ASUR2020-127-S1010

Author Response

Comments received from the reviewer are outlined below, with the author responses in bold.

  1. Lines 415-420: Renovating historic buildings is much better than demolishing them...The question concerning the renovation or demolition of historic or protected buildings is a very serious one. It is not determined exclusively by a factor of cost-effectiveness, which may be secondary in this field. There are many legislative frameworks, such as Italy's, that categorically prohibit the demolition of such existing heritage. Please clarify this approach !
    • It has been clarified that this paper is specifically UK/England-based; we have revised the manuscript to address other factors beyond the challenges of adopting innovations

  1. The article suggests some good perspectives, suggesting some macro-indicators, but it is developed in a purely theoretical form. 
    • This has been added as a limitation of the article to be addressed in further research in the conclusions section.

  1. No connection is made between the approaches in Figure 4 , the LCS approach in Figure 5, as well as the five pillars shown in Figure 8. Nor are relative weights and/or incidences of these parameters determined.
    • We have removed the above images

  1. No clear methodology is defined, nor a model to be tested against one or more case studies (the latter are only described in a purely theoretical way). The bibliography is extensive and relevant, which suggests a thorough analytical phase. But the latter does not translate into concrete methodological elaboration.
    • A brief few sentences are now added to the methodology of the paper. See end of page 2

  1. The following references are recommended: DOI: 10.3390/SU12166432, DOI: 10.3280/ASUR2020-127-S1010
    • Added

Reviewer 4 Report

- The paper refers to the importance of the building sector and the impacts on carbon emissions and climate change. It defines itself “Culture and heritage sectors can also contribute to meeting environmental priorities by reducing their carbon footprint” (line 140) without arguing the quote. The theme is taken up again in paragraph 4 (line 170) without however making some critical arguments in support of the thesis which, in any case, appears to be a priority to the authors so much as to include it also in the abstract;

- In the light of this, it is perhaps necessary in the abstract to better identify the objectives, method and results of the work.

- Arguing better the meaning of Fig 2 and the relationship of the image with the text.

- Paragraph 2. "Global Frameworks and Sustainable Development" tries to provide a framework on the theme of sustainable development and the relationship with the theme of adaptive reuse. The regulatory apparatus on the subject could be updated (especially with respect to emissions and greenhouse gases in the building sector) and the relationship between the arguments supported in the paragraph and the reference to the English situation could be better explained (line 145 )

- Section 3. “The state of planning in England and climate-conscious measures and responses”. In addition to the image, the paragraph have to be argued and suitably commented through a critical reading between the international and the English dimension.

- Lines 190-194: “the heritage sector (including archeology) has begun to address ways in which it can meaningfully respond to (a) understanding its wider role in contributing to the climate emergency, and; (b) recognizing heritage as a powerful asset for climate action”: for a better comprehension please argue this thesis.

The text also fails to give appropriate evidence of how the adaptive reuse of heritage becomes a useful tool for climate control.

- In the text there is a long digression on the meaning and adjective of innovation and, at the end, it appears to introduce the strongly architectural theme of adaptive reuse (line 367). In this regards no further considerations are made.

- Lines 404-407 declares: “Although historical and traditionally built buildings promote resource efficiency, they continue to be stigmatized as "hard to deal with" or energy hungry, despite evidence of several thousand years of proven effectiveness in an environment at low carbon emissions, low environmental energy”: the thesis supported here could be partly true but, for a better understanding, it should also be better argued with other declinations, such as for example functional obsolescence. It would be interesting to understand the authors' critical vision of the built heritage and its potentiality.

- The text cites many English buildings or areas: stables, historical mills, the Tate Modern, King's Cross. To better understand their importance in the paper on the theme of adaptive reuse, it would be advisable to better define the boundaries of interpretation. They are very different buildings, of different values, scale (construction and urban), historical periods: their relationship with the thesis supported in the paper and how they can be considered virtuosos examples on the subject are not clear.

- The conclusions lack a vision of a future and architectonical design perspective and above all of possible actions and strategies to be implemented in order to make the heritage at the service of the environmental problem.

General attention is required in the drafting of the paper: errors make the reading of the document less fluid.

- There are many typing and punctuation errors (line 96, 98, 104, 115, 123, 136, 139, 207, 262, 264, 294, 372, 388, etc….)

Check the numbering of the images carefully (there are two figures 4, two figures 3, there is no fig. 7 and there is the caption of the figure 6 but not the figure itself).

Text errors – page not found (line 427, 437) 

Character style of line 625 is not homogeneous to the others. 

Check the formatting of the references.

Many links to website of the references don't work (page not found): 700 - 705 – 710 – 714 – 717 – 720 – 722 – 733 741 744 – 747 – 752 – 755 – 781 – 784 – 786 – 788 – 792 794 – 797 – 799 – 806 – 819 – 843 – 845 – 862 – 868 – 873 - 876 – 878 – 881 -886 – 888 – 896 – 907 – 914 – 920 – 930 – 935 – 943 – 951 – 953 – 959 – 970 – 977 and 738 page not free.

Author Response

The paper refers to the importance of the building sector and the impacts on carbon emissions and climate change. It defines itself “Culture and heritage sectors can also contribute to meeting environmental priorities by reducing their carbon footprint” (line 140) without arguing the quote. The theme is taken up again in paragraph 4 (line 170) without however making some critical arguments in support of the thesis which, in any case, appears to be a priority to the authors so much as to include it also in the abstract;

In the light of this, it is perhaps necessary in the abstract to better identify the objectives, method and results of the work.

The abstract has been rewritten

Arguing better the meaning of Fig 2 and the relationship of the image with the text.

Figures have been deleted and revised to fit better into the text

Paragraph 2. "Global Frameworks and Sustainable Development" tries to provide a framework on the theme of sustainable development and the relationship with the theme of adaptive reuse. The regulatory apparatus on the subject could be updated (especially with respect to emissions and greenhouse gases in the building sector) and the relationship between the arguments supported in the paragraph and the reference to the English situation could be better explained (line 145 )

This section has been deleted, with more focus on cities from a global action perspective

Section 3. “The state of planning in England and climate-conscious measures and responses”. In addition to the image, the paragraph has to be argued and suitably commented through a critical reading between the international and the English dimension.

Section has been deleted

Lines 190-194: “the heritage sector (including archeology) has begun to address ways in which it can meaningfully respond to (a) understanding its wider role in contributing to the climate emergency, and; (b) recognizing heritage as a powerful asset for climate action”: for a better comprehension please argue this thesis.

This section has been rewritten, and descripted adaptive reuse as one way to do this. The sentence is framing heritage within an international context

The text also fails to give appropriate evidence of how the adaptive reuse of heritage becomes a useful tool for climate control.

I don’t think we have the evidence to answer this question. We do flag that ‘The recent foresight report ‘Twinning the green and digital transitions in the new geopolitical context’ note:

Digital technologies could play a key role in achieving climate neutrality, reducing pollution, and restoring biodiversity. By measuring and controlling inputs, and with increased automation, technologies like robotics and the internet of things could improve resource efficiency and strengthen the flexibility of systems and networks. Energy efficient blockchain-based data management across the lifecycle and value chain of products and services could galvanise the progress towards a more circular economy and competitive sustainability. Digital technologies could also support monitoring, reporting and verification of greenhouse gas emissions for carbon pricing. Digital product passports enable enhanced material, component and end-to-end traceability and make data more accessible, which is essential for viable circular business models. Digital twins could facilitate innovation and the design of more sustainable processes, products, or buildings. (European Commission, 2022a, 2).’

In the text there is a long digression on the meaning and adjective of innovation and, at the end, it appears to introduce the strongly architectural theme of adaptive reuse (line 367). In this regards no further considerations are made.

It was made clearer along the manuscript that adaptive reuse of heritage  is a type of innovation that can contribute to sustainable urban development.  

Lines 404-407 declares: “Although historical and traditionally built buildings promote resource efficiency, they continue to be stigmatized as "hard to deal with" or energy hungry, despite evidence of several thousand years of proven effectiveness in an environment at low carbon emissions, low environmental energy”: the thesis supported here could be partly true but, for a better understanding, it should also be better argued with other declinations, such as for example functional obsolescence. It would be interesting to understand the authors' critical vision of the built heritage and its potentiality.

We don’t have the architectural background to answer this comment.

The text cites many English buildings or areas: stables, historical mills, the Tate Modern, King's Cross. To better understand their importance in the paper on the theme of adaptive reuse, it would be advisable to better define the boundaries of interpretation. They are very different buildings, of different values, scale (construction and urban), historical periods: their relationship with the thesis supported in the paper and how they can be considered virtuosos examples on the subject are not clear.

We wanted to stress the environmental, social and regenerative benefits of the adaptive reuse of different types of buildings. We address in the conclusions that one of the limitations of the paper, that could be overcome with further research, is the application of concrete metrics and targets against which to evaluate the multidimensional nature of the adaptive reuse of heritage. This could be particularly produced for specific types of buildings, scales and historical periods.

The conclusions lack a vision of a future and architectonical design perspective and above all of possible actions and strategies to be implemented in order to make the heritage at the service of the environmental problem.

We don’t have the architectural background to approach this question. However, we emphasize in the manuscript  the importance of questioning that new buildings are the best solution in terms of achieving decarbonization objectives. We suggest different angles to  further explore how heritage can be put at the service of the environmental problem.

General attention is required in the drafting of the paper: errors make the reading of the document less fluid.

This has been addressed

There are many typing and punctuation errors (line 96, 98, 104, 115, 123, 136, 139, 207, 262, 264, 294, 372, 388, etc….)

We cannot unfortunately see these by line (our document does not have numbering), but we have reviewed the document.

Check the numbering of the images carefully (there are two figures 4, two figures 3, there is no fig. 7 and there is the caption of the figure 6 but not the figure itself).

This should be fixed now.

Text errors – page not found (line 427, 437) 

We cannot unfortunately see these by line (our document does not have numbering), but we have reviewed the document.

Character style of line 625 is not homogeneous to the others. 

Are these character style issues dealt with during type-setting?

Check the formatting of the references.

Many links to website of the references don't work (page not found): 700 - 705 – 710 – 714 – 717 – 720 – 722 – 733 741 744 – 747 – 752 – 755 – 781 – 784 – 786 – 788 – 792 794 – 797 – 799 – 806 – 819 – 843 – 845 – 862 – 868 – 873 - 876 – 878 – 881 -886 – 888 – 896 – 907 – 914 – 920 – 930 – 935 – 943 – 951 – 953 – 959 – 970 – 977 and 738 page not free.

This is something we can look at again. As it was not in the previous references, we would have to review this again.

Round 2

Reviewer 2 Report

the paper has improved signficantly. Thanks to the authors who have put quite some efforts into this. My remaining recommendation would be to be a bit clearer on the methods, particularly the body of literature assessed.

Author Response

Hi, 

Thanks for your guidance and feedback. We have added further detail to the methodology. 

Thanks 

Reviewer 3 Report

The recommendations from the first phase of the review were, in main, implemented. The references are up-to-date and relevant.

Author Response

Thanks for your guidance and feedback. Much appreciated.

Reviewer 4 Report

The authors consider punctually all the comments and suggestions cited in the review. They work hard implementing and going deeper on some specific issues cited in the review. For this reason, for my personal view, the paper has been improved and now it can be published as it is

Author Response

Thank you for your guidance and feedback. Much appreciated.